# Guideline Adherence in Antibiotic Prescribing to Patients with Respiratory Diseases in Primary Care: Prevalence and Practice Variation

**DOI:** 10.3390/antibiotics9090571

**Published:** 2020-09-03

**Authors:** Karin Hek, Thamar E.M. van Esch, Anke Lambooij, Yvette M. Weesie, Liset van Dijk

**Affiliations:** 1Nivel, Netherlands Institute for Health Services Research, 3500 BN Utrecht, The Netherlands; thamarvanesch@gmail.com (T.E.M.v.E.); y.weesie@nivel.nl (Y.M.W.); l.vandijk@nivel.nl (L.v.D.); 2IVM, Dutch Institute for Rational Use of Medicine, 3502 GB Utrecht, The Netherlands; a.lambooij@ivm.nl; 3Groningen Research Institute of Pharmacy, Pharmaco Therapy, Epidemiology and Economics, University of Groningen, 9700 AD Groningen, The Netherlands

**Keywords:** antibiotic prescribing, primary care, guideline adherence, respiratory tract infections

## Abstract

Respiratory tract infections (RTIs) account for a large part of antibiotic prescriptions in primary care. However, guidelines advise restrictive antibiotic prescribing for RTIs. Only in certain circumstances, depending on, e.g., comorbidity, are antibiotics indicated. Most studies on guideline adherence do not account for this. We aimed to assess guideline adherence for antibiotic prescribing for RTIs as well as its variation between general practices (GPs), accounting for patient characteristics. We used data from electronic health records of GPs in the Netherlands. We selected patients who consulted their GP for acute cough, rhinitis, rhinosinusitis or sore throat in 2014. For each disease episode we assessed whether, according to the GP guideline, there was an indication for antibiotics, using the patient’s sociodemographic characteristics, comorbidity and co-medication. We assessed antibiotic prescribing for episodes with no or an unsure indication according to the guidelines. We analysed 248,896 episodes. Diagnoses with high rates of antibiotic prescribing when there was no indication include acute tonsillitis (57%), strep throat (56%), acute bronchitis (51%) and acute sinusitis (48%). Prescribing rates vary greatly between diagnoses and practices. Reduction of inappropriate antibiotic prescribing remains a key target to tackle antimicrobial resistance. Insight into reasons for guideline non-adherence may guide successful implementation of the variety of interventions already available for GPs and patients.

## 1. Introduction

Increasing antimicrobial resistance is globally recognized as a major threat to human health [1]. Antimicrobial resistance rates are higher in countries with more antibiotic use and antibiotic prescribing [2]. Acute respiratory tract infections (RTIs) account for a large part of antibiotic prescriptions, whereas evidence of effectiveness is questionable [3,4].

The evidence of effectiveness is collected and summarized in recommendations for physicians in clinical practice guidelines. These guidelines support physicians in their decision on whether or not to prescribe an antibiotic. Guidelines also provide advice for specific patient groups, such as for children, elderly patients or patients with a poor immune response. Most studies assessing antibiotic prescribing do not take into account these patient characteristics which may justify prescription of antibiotics for certain indications [4,5]. An exception is a study by Hope et al. who found that almost 20% of practices among a group of high antibiotic prescribers would not have been classified as high prescribers when taking comorbidity into account [6]. Another exception is the study of Dekker et al. who took into account age, gender, general health state and comorbidity [7]. They showed that even when taking patient characteristics into account, 46% of the antibiotic prescriptions by Dutch GPs for RTIs in 2008–2010 were not indicated by the guidelines. Furthermore, they found that overprescribing was highest for lower RTIs. 

The Netherlands, the setting of this study, has maintained a relatively low antibiotic prescribing in primary care compared with other countries, with a corresponding low antimicrobial resistance rate [2,8]. The Dutch College of General Practitioners (NHG) develops and updates evidence-based guidelines to support Dutch General Practitioners (GPs) in their decision whether or not to prescribe an antibiotic for RTIs. Still, there is considerable variation in antibiotic prescribing for RTIs among Dutch GPs [5]. This practice variation may be caused by differences in patient characteristics between practices that justify differences in antibiotic prescribing. However, it may also be explained by differences in guideline adherence, suggesting prescribing can be improved. The study of practice variation while accounting for patient characteristics is important for uncovering the potential for improvement in antibiotic prescribing and will therefore help to inform and stimulate antimicrobial stewardship.

The aim of the present study was to assess: (i) Guideline adherence to antibiotic prescribing for RTIs in the Netherlands, taking into account patient characteristics, comorbidity and co-medications that may justify prescribing of antibiotics according to guidelines; (ii) practice variation, while accounting for patient characteristics, comorbidity and co-medications.

## 2. Materials and Methods

### 2.1. Nivel Primary Care Database

Data from the Nivel Primary Care Database [9] (Nivel-PCD) were used. Nivel-PCD collects data from routine electronic health records of a large and dynamic pool of general practices across the Netherlands over time. Data comprise information on patient characteristics, consultations, morbidity, prescriptions and lab test results. Morbidity is recorded in each consultation using the International Classification of Primary Care version 1 (ICPC-codes) [10]. Consultations were grouped into illness episodes [11]. Registrations of complaints (ICPC-codes R01 to R29, such as cough and sneezing) were combined into episodes of diseases (R70 to R99, such as upper respiratory infection) if both were registered. Complaints were only analysed separately if there was no matching disease registered. Prescription data were recorded using the Anatomical Therapeutic and Chemical (ATC) classification. We used electronic health records data from 2014. These included patients of 307 practices (with in total 1,164,808 listed patients) from which we selected patients with relevant illness episodes (see below). Data protection is described in Box 1. 

Box 1Data protection Nivel-PCD.Dutch law allows the use of extracts of electronic health records for research purposes under certain conditions. According to Dutch legislation, neither obtaining informed consent nor approval by a medical ethics committee is obligatory for this kind of observational study containing no directly identifiable data [12]. With respect to Nivel-PCD, participating general practices are contractually obliged to inform their patients about their participation in Nivel-PCD and to inform patients about the possibility to opt-out if they object to their data being included in the database. This study has been approved by the applicable governance bodies of Nivel-PCD under nr. NZR-00315.066.

### 2.2. Study Sample

The study sample consisted of the 206,473 registered patients from the 307 selected general practices (136 to 3247 per practice), who, according to Nivel-PCD, contacted their GP in 2014 for: acute cough (recorded using ICPC-codes: R05—acute cough; R71—whooping cough; R77—laryngitis/trachitis; and R78—acute bronchitis/bronchiolitis);allergic and non-allergic rhinitis (recorded using ICPC-codes: R07—sneezing/nasal congestion; R08—nose symptom/complaint other; and R97—allergic rhinitis),acute rhinosinusitis (recorded using ICPC-codes: R09—sinus symptom/complaint; R74—upper respiratory infection acute; and R75—sinusitis acute/chronic)acute sore throat (recorded using: R21—throat symptom/complaint; R22—tonsils symptom/complaint; R72—strep throat/scarlet fever; R76—tonsillitis acute).

We excluded episodes with an ICPC-code for which an antibiotic is indicated according to the guideline (i.e., pneumonia). 

### 2.3. Measurements

#### 2.3.1. Antibiotics Indication 

For the included diagnoses, the indication for antibiotics depends on, for example, the patient’s age and comorbidity (according to Dutch clinical guidelines developed by the NHG, the NHG guidelines [13,14,15,16]). We therefore evaluated prescribing antibiotics separately for (i) episodes with an unsure indication for antibiotics according to the guideline (antibiotics could be considered) and (ii) episodes for which antibiotics are not indicated according to the guideline (antibiotics should generally not be prescribed). We determined whether an antibiotic was (possibly) indicated for each episode of illness according to guideline recommendations using the patients’ age, morbidity data, lab test results and prescription data from Nivel-PCD. Table 1 provides an overview of all relevant guideline recommendations and definitions that were used to estimate the indication for antibiotics.

#### 2.3.2. Antibiotics Prescriptions (Dependent Variable)

We selected episodes in which an antibiotic was not indicated or with an unsure indication according to the guidelines (see above). For each selected episode of illness, we assessed whether or not an antibiotic (ATC subgroup J01) had been prescribed using Nivel-PCD prescription data, linking prescriptions to episodes based on dates and ICPC codes. This outcome (antibiotics: Yes/no) was used as the dependent variable. 

### 2.4. Statistical Analyses

First, descriptive statistical analyses were performed in order to gain insight into numbers of relevant illness episodes, the indication for antibiotics and the GP’s prescription of antibiotics. Next, logistic multilevel regression analyses were performed to estimate practice variation. For each ICPC code, and separately for episodes with no or an unsure antibiotic indication, we assessed the practice variation in antibiotic prescribing. The models contained two levels, as the data are hierarchically structured with patients nested in general practices. A multilevel analysis takes into account the nested structure of the data as well as the differences in the number of patients per practice. As the majority of patients (87%) had one episode per ICPC, we only analysed the first episode per patient per ICPC. We included the GP’s prescription of antibiotics as outcome variable (an antibiotic prescribed yes or no), and patient’s age and sex as control variables. The level of significance was set at 0.05. All analyses were performed using STATA, version 14.0.

## 3. Results

We included 248,896 RTI episodes with no or an unsure indication for antibiotics according to the guidelines of 206,473 unique patients. Acute upper respiratory tract infections and cough had the highest prevalence (63,004 and 52,285 episodes, respectively, see Table 2). Tonsil complaints and whooping cough had the lowest prevalence (1487 and 666 episodes, respectively). According to NHG guidelines there was no indication for antibiotics for 62% (45,134) of acute cough episodes, 100% (53,907) of allergic and non-allergic rhinitis episodes, 87% (65,253) of the sinusitis episodes and 87% (30,011) of the sore throat episodes. 

### 3.1. Antibiotic Prescribing

Figure 1 shows the percentage of episodes with an antibiotic prescription for each RTI, separately for episodes without an antibiotic indication and for episodes with an unsure antibiotic indication (antibiotics might be considered). Antibiotic prescribing for episodes without antibiotics indication was highest for diagnoses described in the guidelines “Sore throat”. Both strep throat and acute tonsillitis had a high antibiotic prescribing rate (56% and 57%). The antibiotic prescribing rate was highest for strep throat episodes with an unsure indication (64%). For diagnoses described in the guideline “Acute cough”, the antibiotic prescribing rate was highest for acute bronchitis/bronchiolitis. In 51% of the episodes with an unsure indication an antibiotic was prescribed. Moreover, in 51% of acute bronchitis episodes without an indication an antibiotic was prescribed. For the diagnoses in the guideline “Sinusitis” antibiotics were frequently prescribed also in cases where there was no indication (48% in case of acute sinusitis). Hardly any antibiotics were prescribed for diagnoses described in the guideline “Rhinitis”. 

### 3.2. Practice Variation 

Table 2 illustrates the variation between general practices in antibiotic prescribing for RTIs with an unsure and without antibiotic indication. Amongst RTIs without an indication for antibiotics the widest 95% practice range was observed for acute bronchitis/bronchiolitis (18–83%), followed by strep throat (24–83%), acute/chronic sinusitis (23–75%) and acute tonsillitis (32–79%). For RTIs where antibiotics could be considered the widest practice range was observed for whooping cough (1–83%), followed by acute laryngitis/tracheitis (3–84%), sinus symptoms and complaints (2–66%), acute bronchitis/bronchiolitis (23–79%) and acute/chronic sinusitis (24–74%).

## 4. Discussion 

In this study, we found that adherence to guidelines for rhinitis was good, as hardly any antibiotics were prescribed for rhinitis and rhinitis associated diagnoses. However, half of the cases with bronchitis, sinusitis, strep throat and tonsillitis were treated with an antibiotic, whereas there was no indication for this in the guidelines. Furthermore, we observed large inter-practice variation in situations where there was no indication for an antibiotic. This variation between practices could not be explained by differences between patient populations that may justify the prescription of an antibiotic, such as comorbidity, patient age, co-medication and lab results. 

The highest percentages of antibiotic prescribing were found for RTIs related to the guideline acute sore throat, such as acute tonsillitis and strep throat, but also for acute bronchitis (guideline acute cough). This is in line with other European studies [5,7,17,18,19]. In the current study we assessed for each studied episode of RTIs whether, according to the guidelines, antibiotics could be considered or not. A previous study by Hope et al. showed the importance of taking into account factors such as comorbidity, like we did in the current study, to determine guideline adherence [6]. The high percentage of antibiotic prescribing in episodes for which we could not find an indication for antibiotics suggests that guideline adherence can be improved.

Moreover the wide 95% practice range suggests that there is room for improvement, although antibiotic prescribing to RTIs in the Netherlands is generally lower than in other countries. Over the past years many interventions to reduce inappropriate prescribing of antibiotics for RTIs have been developed, tested and evaluated [20,21]. Recent reviews suggest that multifaceted interventions directed at for example educational interventions for both patients and clinicians, procalcitonin testing in adults and electronic decision support work best in reducing inappropriate antibiotic prescribing [22]. A survey in England among primary care stakeholders suggested among others peer learning and individual prescriber audits as ways to improve antimicrobial stewardship programs [23]. In the Netherlands, parts of these interventions have been implemented. However, the large practice variation suggests that the implementation of guidelines, decision support tools and other interventions to reduce antibiotic prescribing for RTIs can be further improved. The methodology used in the current study, provides further precision to antibiotics indicators, that could be used in individual prescriber audits. 

The study of practice variation helps to pinpoint practices with high antibiotic prescribing to target interventions as well as practices with low antibiotic prescribing. This allows to find out which modifiable factors are associated with low and high antibiotic prescribing, such as the level of shared decision making that GPs use [24] or the rate of diagnostic testing [25]. The outcome of such a study could be used in the design of future interventions and in antimicrobial stewardship programs. 

Notably, we found that antibiotic prescribing was nearly as high in situations where antibiotics could be considered according to the guideline as in situations with no indication for antibiotics. This may indicate that additional patient related factors, that were beyond the scope of this study, play an important role in the decision to prescribe an antibiotic. An example of such factor may be patient preference, or the GP’s perception of the patient’s preference [26,27,28,29]. 

## 5. Strengths and Limitations

The main strength of the current study is that we used data from a large nationally representative database from which antibiotic prescribing as well as relevant patient characteristics such as comorbidities, co-medication and CRP measurements could be retrieved. 

Using a registration database has the downside that the results depend on the accuracy and completeness of coding by the participating GPs. Differences in coding between GPs may partly explain the large practice variation between GPs we observed. Some GPs may for example code bronchitis (diagnosis) as acute cough (symptom). This may affect antibiotic prescribing rates in these practices. However, nationally there has been a lot of attention for completeness and quality of coding by GPs, and this resulted in improved diagnostic coding in the years before this study, so we do not expect this to have a large impact on our results [30].

While we were able to assess many patient characteristics, we had no information on disease severity and prolonged or recurring fever. According to guidelines, these factors may justify the prescribing of antibiotics, even in patients and for diagnoses for which antibiotics are usually not recommended. As a result disease severity and fever may greatly influence the decision to treat an RTI with an antibiotic. This implies that we overestimated the percentage of antibiotic treatment in RTIs without an antibiotic indication. However, we do not expect the percentage of patients with severe disease or fever to differ greatly between practices, so this does not explain the wide 95% practice range. Indeed, a study by Stuart et al. showed that the difference in patient’s illness severity scores account for only a minority of RTI antibiotic prescribing variation between general practices [31].

The current study focused on practice variation in antibiotic treatment for RTIs in which antibiotics are not indicated or possibly indicated. We did not assess antibiotic prescribing for RTIs for which antibiotics are indicated, such as pneumonia. A study on guideline adherence in RTIs for which antibiotics are indicated may reveal cases of undertreatment with antibiotics. Furthermore we did not assess symptom relief or other outcomes of patients who did or did not receive an antibiotic. Therefore we cannot conclude whether the great observed practice variation led to different patient outcomes.

## 6. Conclusions and Implications

The current study suggests that, although antibiotic prescribing for RTIs in the Netherlands is low compared to many other countries, improvement is possible and warranted particularly for bronchitis, sinusitis, strep throat and acute tonsillitis. Reduction of inappropriate antibiotic prescribing remains a key target to tackle the global threat of antimicrobial resistance. To further improve guideline adherence, insight into reasons for non-adherence is required to guide improvement and successful implementation of the variety of interventions that are already available for GPs and patients. We can benefit from practice variation by studying differences between high and low antibiotic prescribers. 

## Figures and Tables

**Figure 1 antibiotics-09-00571-f001:**
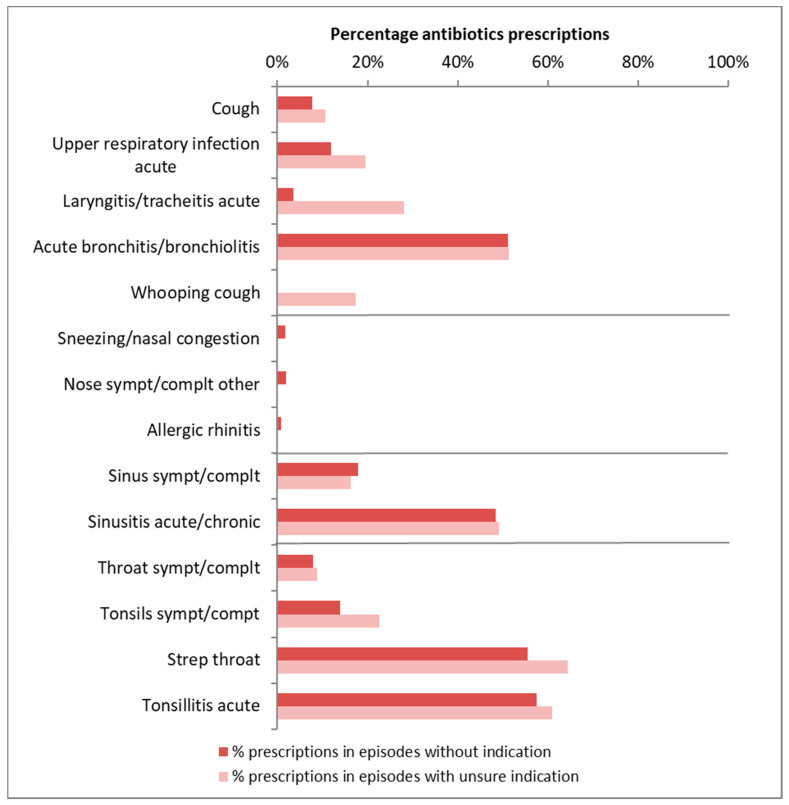
Percentage antibiotics prescription per RTI with unsure and without antibiotic indication.

**Table 1 antibiotics-09-00571-t001:** Recommendations for prescribing of antibiotics for acute cough, allergic and non-allergic rhinitis, acute rhinosinusitis and acute sore throat and study definitions.

	Guideline			
	Acute Cough, (M78, 2011) [14]	Allergic and non-Allergic Rhinitis (M48, 2006) [16]	Acute Rhinosinusitis, (M33, 2014) [14]	Acute Sore Throat (M11, 2015) [15]
**Diagnoses included in the guideline (ICPC)**	Acute cough (R05), Whooping cough (R71), Laryngitis/tracheitis acute (R77), Acute bronchitis/bronchiolitis (R78)	Sneezing/nasal congestion (R07), Nose symptom/complaint other (R08) Allergic rhinitis (R97)	Sinus symptom/complaint (R09), Upper respiratory infection acute (R74), Sinusitis acute/chronic (R75)	Throat symptom/complaint (R21), Tonsils symptom/complaint (R22), Strep throat/scarlet fever (R72), Tonsillitis acute (R76)
**Antibiotics recommen-dations in guideline**	No antibiotics if pneumonia is not considered likely. Exceptions in which antibiotics should be considered are patients with one or more risk factors: • Age < 3 months or > 75 years • Relevant comorbidity: heart failure, severe COPD, diabetes mellitus (in particular when using insulin), neurological diseases, severe kidney diseases. • Poor immune response • CRP in adults: <20 mg/L no indication for antibiotics, 20–100 mg/L indication for antibiotics depends on the clinical presentation, >100 mg/L indication for antibiotics.	Antibiotics are not mentioned in the guideline.	In principle, no antibiotics. Antibiotics are indicated in patients who are seriously ill. Antibiotics can be considered in patients with poor immune response: • Chronic use of corticosteroids or other immunosuppressive medicines • HIV infection with a reduced number of T-cells • Chemotherapy or radiotherapy • Immune disorders • Frail elderly who are sick • Patients with diabetes mellitus Antibiotics can be considered for patients who have had fever for more than 5 days, or for patients who have recurrent fever after a few fever-free days within one episode of rhinosinusitis.	In principle, no antibiotics. Antibiotics are indicated • in seriously ill patients • if advised by public health services in the rare case of scarlet fever clusters in a closed community. Antibiotics can be considered in patients with an increased risk of complications, e.g., in case of: • Chronic use of corticosteroids or other immunosuppressive medicines • HIV infection with a reduced number of T-cells • Chemotherapy or radiotherapy • Cancer • Immune disorders • Diabetes mellitus • Rheumatic fever • Severe alcohol abuse • Drug abuse • Functional asplenic • sickle cell disease
**Study definitions Antibiotics not indicated**	in patients with cough (R05, R77, R78) between three months and 75 years, without indications for poor immune response *, with CRP <20 and without relevant comorbidity.	in all patients	in patients with sinus complaints (R09, R74, R75) without indications for poor immune response *.	in patients with sore throat complaints (R21, R22, R72, R76) without indications for poor immune response * and without rheumatic fever in their medical history.
**Study definitions** **Antibiotics possibly indicated**	in patients with cough (R05, R77, R78) younger than three months or over 75 years, or with indications for poor immune response *, or with CRP >20 or with relevant comorbidity and in patients with whooping cough (R71).	not applicable	in patients with sinus complaints (R09, R74, R75) with an indication for poor immune response *.	in patients with sore throat complaints (R21, R22, R72, R76) with an indication for poor immune response * or with rheumatic fever in their medical history.
**Study definitions** **Remarks**	Not all measured CRP values are recorded. CRP limits for indications are only applied if CRP values were recorded. Relevant comorbidity includes: heart failure, COPD, neurological diseases and severe kidney diseases.	not applicable	Being seriously ill and having prolonged or recurrent fever cannot be retrieved from Nivel Primary Care Database and are consequently not taken into account.	Being seriously ill cannot be retrieved from Nivel Primary Care Database and is consequently not taken into account. The same holds for scarlet fever clusters in a closed community.

* Patients are considered as having a poor immune response if at least one of the following drugs were prescribed as described in the guideline: Corticosteroids (chronic use), cytostatic drugs, DMARDs, biologicals, anti-thyroid drugs, phenytoin, neuroleptics, antivirals for systemic use or if at least one of the following diseases was recorded: HIV infection, cancer, diabetes mellitus, severe alcohol abuse, sickle cell disease, (functional) asplenic, severe renal insufficiency.

**Table 2 antibiotics-09-00571-t002:** Practice variation in antibiotic prescribing for respiratory tract infections (RTIs) with unsure and without antibiotic indication.

ICPC	Antibiotics Indicated	Number of Episodes	Number of Practices	Mean % with AB Prescription	% with AB Prescription (95% Range)
**Guideline acute cough**					
Cough (R05)	no	33,571	307	8%	2–25%
	unsure	18,714	307	11%	3–29%
Upper respiratory infection acute (R74)	no	45,717	307	12%	3–36%
	unsure	17,287	307	20%	6–48%
Laryngitis/tracheitis acute (R77)	no	1653	283	4%	0–27%
	unsure	484	158	28%	3–84%
Acute bronchitis/bronchiolitis (R78)	no	9910	306	51%	18–83%
	unsure	8176	307	51%	23–79%
Whooping cough (R71)	unsure	666	190	17%	1–83%
**Guideline allergic and non allergic rhinitis**					
Sneezing/nasal congestion (R07)	no	5331	303	2%	0–6%
Nose sympt/complt other (R08)	no	5977	306	2%	1–6%
Allergic rhinitis (R97)	no	42,599	307	1%	0–2%
**Guideline acute rhinosinusitis**					
Sinus sympt/complt (R09)	no	2176	285	18%	4–51%
	unsure	532	182	16%	2–66%
Sinusitis acute/chronic (R75)	no	17,360	307	48%	23–75%
	unsure	4316	306	49%	24–74%
**Guideline sore throat**					
Throat sympt/complt (R21)	no	17,889	307	8%	2–29%
	unsure	3549	307	9%	2–32%
Tonsils sympt/compt (R22)	no	1368	283	14%	3–50%
	unsure	119		23%	-
Strep throat/scarlet fever (R72)	no	1457	259	56%	24–83%
	unsure	166		64%	-
Tonsillitis acute (R76)	no	9297	307	57%	32–79%
	unsure	582	210	61%	36–81%

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
