# Peer review of "Guideline Adherence in Antibiotic Prescribing to Patients with Respiratory Diseases in Primary Care: Prevalence and Practice Variation"

_antibiotics, 2020, doi:10.3390/antibiotics9090571_

Round 1

Reviewer 1 Report

The topic is very interesting and the study was done in Netherland that has great standard of infection control and antibiotic stewardship. However: 1) The use of the definition "unsure indication" of antibiotic therapy is too vague. Unfortunately, each situation of "unsure indication" is highly questionable and this significantly influences the results of the study as well as their interpretation 2) Authors define the object of the study "RTIs" (respiratory tract infections) but they did not include pneumonia. This limitation should be discussed or, in alternative, RTI should be modified in "upper respiratory infections" 3) Table with antibiotic indication needs to be more clear, separating Indications to antibiotics (with more references added) from No indication to antibiotic therapy 4) Reasons for which several drugs are linked to "poor immune response" should be discussed (line 121) 5) Strep throat should be better defined along the paper and a discussion on antibiotic indications should be provided. In particular, when Strep throat has "unsure indications"?

Author Response

We thank the reviewers for their valuable comments that helped us to improve our manuscript. We provide a point-by-point reply below.

The topic is very interesting and the study was done in Netherland that has great standard of infection control and antibiotic stewardship.

However:

1) The use of the definition "unsure indication" of antibiotic therapy is too vague. Unfortunately, each situation of "unsure indication" is highly questionable and this significantly influences the results of the study as well as their interpretation

Authors’ reply: With unsure indication for antibiotics we refer to diagnoses for which antibiotics could be considered according to the guidelines of the Dutch College of General Practitioners (NHG), yet where the guideline does not provide a straightforward advise on whether or not to prescribe an antibiotic. This is for instance the case in patients with sinus complaints (R09, R74, R75) with an indication for poor immune response. In these cases shared decision making of GP and patient plays an important role. We believe it is important to report this group separately from those where the guideline is straightforward in prescribing no antibiotic as antibiotic prescribing rates and variation between practices may be different in his group. This is now explicitly mentioned in the abstract, method, results and discussion section.

2) Authors define the object of the study "RTIs" (respiratory tract infections) but they did not include pneumonia. This limitation should be discussed or, in alternative, RTI should be modified in "upper respiratory infections"

Authors’ reply: Indeed, we did not study diagnoses like pneumonia for which antibiotics are always indicated. We were interested in diagnoses for which antibiotics are not always indicated and where there is more room for interpretation on whether or not to prescribe antibiotics. This is mentioned in the methods section: “We excluded episodes with an ICPC-code for which an antibiotic is indicated according to the guideline (i.e. pneumonia).” As lower RTI bronchitis is also included in the study, we prefer to keep the term RTI. The limitation is now mentioned in the discussion.

Changes to the manuscript: We added the following paragraph to the discussion: The current study focused on practice variation in antibiotic treatment for RTIs in which antibiotics are not indicated or possibly indicated. We did not assess antibiotics prescribing for RTIs for which antibiotics are indicated, such as pneumonia. A study on guideline adherence in RTIs for which antibiotics are indicated may reveal cases of undertreatment with antibiotics.

3) Table with antibiotic indication needs to be more clear, separating Indications to antibiotics (with more references added) from No indication to antibiotic therapy

Authors’ reply: References to the guidelines are in the header of the table. Each of the guidelines advises on more than one diagnosis, e.g. the guideline acute cough not only gives treatment advise for the diagnosis acute cough, but also for whooping cough (R71), laryngitis / tracheitis acute (R77), and acute bronchitis / bronchiolitis (R78). The table describes the advise in the guideline and separately the operationalisation of guideline advise for situations with no or an unsure indication of antibiotics.

Changes to the manuscript: We specifically mention that the table includes study definitions in the table title. Also we slightly adapted the table in order to make it more clear.

4) Reasons for which several drugs are linked to "poor immune response" should be discussed (line 121)

Authors’ reply: poor immune response was defined as was mentioned in the guidelines, this is now explicitly mentioned in table 1.

5) Strep throat should be better defined along the paper and a discussion on antibiotic indications should be provided. In particular, when Strep throat has "unsure indications"?

Author´s reply: ICPC-code R72 includes strep throat and scarlet fever. Like for other acute sore throat complaints antibiotics are generally not advised for strep throat in Dutch guidelines. Antibiotics may be considered in seriously ill patients, patients with increased risk of complications and in case of scarlet fever clusters (see table 1). These indications were extracted from the Dutch GP guidelines for acute sore throat, including strep throat. We included the description of scarlet fever to the method section and table 1.

Reviewer 2 Report

Hek et al analyzed NIVEL data for antibiotics prescriptions for respiratory diseases. Analysis showed that increased antibiotic prescription where a clear suggestion of antibiotic class (unsure indication) or no suggestion of antibiotics(no indication) is given in GP guidelines. These prescription serves as general practitioners practice variations. Overall study highlights the increased prescription of antibiotics and is important for guiding clinical interventions for antibiotic usage.

Study would benefit more if author can address the following:

  1. In cases where antibiotic prescription lead to relief from respiratory system vs where no antibiotics were given. As currently it is unclear that how much of these prescriptions were a success.

Author Response

We thank the reviewers for their valuable comments that helped us to improve our manuscript. We provide a point-by-point reply below.

Hek et al analyzed NIVEL data for antibiotics prescriptions for respiratory diseases. Analysis showed that increased antibiotic prescription where a clear suggestion of antibiotic class (unsure indication) or no suggestion of antibiotics(no indication) is given in GP guidelines. These prescription serves as general practitioners practice variations. Overall study highlights the increased prescription of antibiotics and is important for guiding clinical interventions for antibiotic usage.

Study would benefit more if author can address the following:

In cases where antibiotic prescription lead to relief from respiratory system vs where no antibiotics were given. As currently it is unclear that how much of these prescriptions were a success.

Authors’reply: The reviewer suggests to assess outcomes of antibiotic prescriptions in comparison to no antibiotic prescription. This is indeed very interesting, but was not the aim of the current study in which we assessed guideline adherence. We added a sentence to the discussion describing this as an important research topic for future research .

Changes to the manuscript: we added the following paragraph to the discussion: Furthermore we did not assess symptom relief or other outcomes of patients who did or did not receive an antibiotic. Therefore we cannot conclude whether the great observed practice variation led to different patient outcomes.